# Triggerable Super Absorbent Polymers for Coating Debonding Applications

**DOI:** 10.3390/polym13091432

**Published:** 2021-04-29

**Authors:** Ioannis A. Kartsonakis, Panagiotis Goulis, Costas A. Charitidis

**Affiliations:** Research Unit of Advanced, Composite, Nano-Materials and Nanotechnology, School of Chemical Engineering, National Technical University of Athens, 9 Heroon Polytechniou St., Zographos, GR-15773 Athens, Greece; pgoulis@chemeng.ntua.gr

**Keywords:** coatings, SAPs, steam, debonding, triggerable polymers

## Abstract

This study aims to examine how core–shell super absorbent polymers (SAPs) can be effective in relation to recycling processes by using them as triggerable materials in coating binders. Super absorbent polymers are partially cross-linked, three-dimensional polymer networks that can absorb and retain water. Coatings based on an acrylic binder, including SAPs, were applied onto plastic substrates of acrylonitrile–butadiene–styrene/polycarbonate. The incorporation of 1 wt.% and 5 wt.% SAPs into the coatings resulted in the debonding of the coatings from the substrates under a steam treatment. The trigger mechanism for the core–shell hydrophilic SAPs relies on the different abilities of the core and shell materials to be swollen. Therefore, under the influence of steam, SAPs can enhance their shape due to water absorption and the breaking of the inorganic shell. This results in the reduction of the attachment between the primer layer and both the top coating and the substrate, thus enabling the detachment of the top coating from the corresponding substrate. The obtained results from this study can be considered as potential formulations for plastic recycling applications in industries.

## 1. Introduction

The recycling of coatings and substrates has attracted a lot of scientific interest lately because the need to reuse and better take advantage of these types of materials is now more requisite than ever [1,2,3]. Coatings are currently used in many scientific applications, such as extending the overall lifetime of materials, protecting against corrosion, waterproofing, and insulating. Furthermore, many industrial experimental paths involve the deposition of a thin film of a functional material to a substrate, which is usually plastic, textile, paper, or foil [3,4]. The functional material is often applied to the substrate in liquid form, or as either a gas or solid. The problem with these processes is that it has proven to be considerably difficult to find a successful and effective way to make all the compounds involved in them reusable [5].

A recent investigation of the recyclability of powder coatings showed that 100% recyclability can never be achieved in reality [6,7]. However, there is a real need to increase this percentage in the years to come as much as possible to ensure that when materials reach their end of life, there is a reasonable possibility for them to be reused in the manufacture of similar final products. Additionally, concerning the reusability of such materials, some attempts were made to remove coatings from polymer substrates with solid particle blasting. Although it was found that paint could be removed from the substrates, a very large dose of particles was required. The high compliance of both the coating and the substrate resulted in difficulty with removing the paint films [2].

Moreover, because the low efficiency of current coating-removal technologies has led to excessive energy consumption, it is of great importance to achieve energy conservation in order to relieve the environmental impact caused by sustainable development, as stated in the study of Zha, et al. [8]. Moreover, in the work of Yun, et al. [9], it was shown that if the energy consumption is optimized, then coatings can be removed using a hybrid laser–waterjet process for recycling scrap coated and cemented carbide tools. Reducing the energy consumption during processing could significantly improve the environmental performance of manufacturing systems.

The indentation technique using a conical or spherical indenter is another idea for the debonding of a polymer coating from the substrate [10,11]. However, the debonding—except when used for recycling—can also have different effects, such as the hybrid progressive degradation of polymer composites [12,13]. It was also recently proven that nanoscale interface debonding and multimode fracture in polymer carbon composites have a correlation with long-term hygrothermal effects [14,15].

Nonetheless, the most promising way to achieve effective separation and reusability of coatings and substrates is by incorporating adhesives and additives into the binders (from resin or other materials), which are used to form the coating [16,17]. Especially regarding plastics and textiles, several additives can give them new life by aiding with the debonding of coatings that are difficult to recycle [18]. Additionally, by modifying certain properties of the polymeric coatings, such as crystallinity and hydrogen bonding, these adhesives can help optimize adhesion strength at low bonding temperatures [19]. Core–shell structures can be fabricated using a wide variety of methods, such as electrospinning (Shao, et al.) [20], electrospraying (Wang, et al.) [21], molecular self-assembly (Simonova, et al.) [22], and the usage of micelles as templates (Kiss, et al.) [23].

In this work, triggerable core–shell SAPs were synthesized with the idea of incorporating them into coating binders. After the enriched binder’s deposition onto acrylonitrile–butadiene–styrene/polycarbonate substrates (ABS/PC) and ABS substrates, the SAPs were triggered by taking advantage of their water absorbance property. Steam was applied onto the coating surface, which caused the additives to swell and become inflated. Subsequently, this effect resulted in the formation of cracks, which in turn separated the coating from the substrate and completed the debonding process. This experimental path constitutes a novel alternative to modern technologies and encourages the use of polymers for advanced technologies and applications.

## 2. Materials and Methods

### 2.1. Materials

Potassium persulfate (KPS, initiator, Sigma Aldrich, St. Louis, MO, USA); acetonitrile (ACN, Sigma Aldrich, St. Louis, MO, USA); ammonium hydroxide (NH_4_OH, Sigma Aldrich, St. Louis, MO, USA); tetraethyl orthosilicate (TEOS, Sigma Aldrich, St. Louis, MO, USA); ethylene glycol dimethacrylate (EGDMA, Sigma Aldrich, St. Louis, MO, USA); Edolan AB binder (aqueous dispersion of an acrylic polymer, Tanatex Chemicals, CENTEXBEL); acrylonitrile–butadiene–styrene (ABS, MAIER); and acrylonitrile–butadiene–styrene/polycarbonate plastic plaque substrates (ABS/PC, MAIER) were used as received, without any further purification. Methacrylic acid (MAA, Sigma Aldrich, St. Louis, MO, USA) was double distilled under vacuum prior to use.

### 2.2. Synthesis of Super Absorbent Polymers

The core–shell SAPs were fabricated according to the following procedure. Radical polymerization was performed to synthesize SAPs consisting of copolymers of P(MAA-*co*-EGDMA). The synthetic process is described in more detail in our previous work [24]. The polymerization took place in a reactor equipped with a condenser under an inert atmosphere (nitrogen, N_2_, flow), at 90 °C, with constant and vigorous stirring. The solvent ACN was added in the reactor vessel under vigorous stirring. When the temperature reached the set point, the distilled MAA and the EGDMA were dropwise supplied to the reactor. Finally, the initiator (KPS) was added to the reaction mixture in the same manner, and the polymerization reaction continued overnight (Figure 1). The copolymer was centrifuged and then washed with ACN twice. The produced P(MAA-*co*-EGDMA) material was stored as a paste in ACN. It was subsequently modified according to the following process.

The P(MAA-co-EGDMA) paste was diluted in ACN under vigorous stirring. TEOS was subsequently added dropwise to the mixture. The reaction was left to proceed overnight, after which the product was centrifuged and then washed with ACN. The hybrid organic–inorganic core–shell P(MAA-co-EGDMA)@SiO_2_ SAPs were stored as a paste in ACN. Additionally, it is worth mentioning that a new alternative method for creating core–shell polymeric structures, which is rapidly gaining interest, is electrospinning [25]. Moreover, the alcohol-thermal technique can be used in the fabrication of core–shell inorganic metastable intermolecular composites, as discussed in the work of Zhou, et al. [26]. In the study of Yusuf, et al. [27], core–shell structures of visible-light-harvesting materials were produced through a one-pot hydrothermal technique.

The produced SAPs were characterized in respect to their morphology, composition, and structure. The absorption ratio for all SAPs was determined in distilled water at two different pH values, 5.5 ± 0.1 and 10.0 ± 0.1. An evaluation of the absorption ratio in both acidic and basic pH was performed in order to determine if both types of water can be used with the Steam Mop Deluxe + Steambuster instruments for the steam treatment of the coatings.

### 2.3. Synthesis of Coatings

The coatings were fabricated by applying a mixture of 10 g of Edolan AB primer—including either 0.1 g or 0.5 g of SAPs—onto plastic plaques ABS and ABS/PC substrates, using a 50 μm BYK coating roller applicator. It is worth mentioning that Edolan AB composition is not available to the public. Edolan AB primer is a white aqueous dispersion of an anionic acrylic polymer with a density of 1.0–1.1 g/cm³ that is miscible with water. It consists of C, O, Al, Si, and Ti elements as evidenced by the EDS analysis. After the application, the coatings were left to dry in room temperature for about 20 min and then further dried at 80 °C for 3 h in an oven. The obtained coatings were treated with steam for the debonding to take place. The steam was applied to both the coated substrate’s surface and the scraped coated substrate’s surface for 5 min in order to test the effectiveness of the debonding.

### 2.4. Characterization Methods

Scanning electron microscopy (SEM) was utilized to characterize the morphology and size of the SAPs and the coatings prior to and after the steam treatment. A Phillips Quanta Inspect (FEI Company) microscope—QUANTA 200 with W (tungsten) filament 25 kV equipped with EDAX GENESIS (AMETEX PROCESS & ANALYTICAL INSTRUMENTS)—was used for the SEM characterization. All the samples were coated with gold before the SEM observation. The core–shell structure of the SAPs was evaluated and confirmed by high-resolution transmission electron microscopy (HRTEM) through a JEM2000 FX (200 KV, resolution 0.28 nm).

The elemental composition of the inorganic SiO_2_ shell of the core–shell SAPs was characterized by energy dispersive X-ray analysis (EDS) using the aforementioned EDAX GENESIS equipment. Thermogravimetric analysis (TGA) (STA 449 F5 Jupiter and Proteus Software, NETZSCH) was also performed in an inert atmosphere and in synthetic air (80% N_2_, 20% O_2_), with a gas flow of 50 mL/min and with a heating rate of 5 °C/min.

To determine the absorption ratio (AR) of SAPs (gg^−1^), weighed amounts of SAPs (M1) and excess amounts of distilled water were added to tubes. At specific time intervals (5, 10, 15, 20, 30, 60, 360, 720, and 1440 min), the SAPs were centrifuged at 15,000 rpm for 5 min, the supernatant liquid was removed, and the SAPs were weighed again (M2). The absorption ratio, AR(%), was calculated using Equation (1). The distilled water had a pH value of 5.5 ± 0.1 due to the hydrolysis of the absorbed CO_2_ and the formation of H^+^ and HCO_3_^−^. The solution with a pH value of 10.0 ± 0.1 was obtained using NH_4_OH.
(1)$$\mathrm{AR}\left(\%\right)=\frac{\left(M_{2}-M_{1}\right)}{M_{1}}\;\times \;100,$$

The steam treatment of the coatings was performed with a Steam Mop Deluxe + Steambuster instrument (Black&Decker).

## 3. Results and Discussion

### 3.1. Evaluation of Super Absorbent Polymers

The SEM image of the produced P(MAA-co-EGDMA) SAPs is illustrated in Figure 2a. It can be observed that spheres were produced according to the aforementioned synthetic process. The size of the obtained SAPs ranged from 170 to 210 nm. The EDS analysis of the P(MAA-co-EGDMA) SAPs revealed that the SAPs consist only of oxygen and carbon due to the organic copolymers. The presence of Au is attributed to the gold plating process, while the very low value of potassium (within the error limit) is attributed to the initiator (Figure 2b).

Furthermore, considering the SEM image of the P(MAA-co-EGDMA)@SiO_2_ core–shell SAPs, it can be seen that they are spheres, and that their diameter is 310 ± 50 nm (Figure 3a). Taking into account the TEM image, the core–shell structure of the SAPs is clearly confirmed (Figure 3b). According to the EDS analysis, they consist of carbon due to the organic polymeric core, of silicon due to the inorganic shell, and of oxygen, which is attributed to both the core and the shell (Figure 3c).

Figure 4 illustrates the FTIR spectra of the produced (a) P(MAA-co-EGDMA) and (b) P(MAA-co-EGDMA)@SiO_2_ SAPs. These spectra confirmed the formation of the inorganic shell on the polymeric core. More specifically, both spectra of P(MAA-co-EGDMA) and P(MAA-co-EGDMA)@SiO_2_ reveal broad bands at 3200–3400 cm^−1^, which are attributed to the -OH stretching vibration due to ambient water absorbance. Furthermore, in Figure 4b, the peaks at 1046 cm^−1^, 782 cm^−1^, and 445 cm^−1^ are attributed to amorphous Si-O-Si vibrations [7], while the absorbance vibrations from 1020 cm^−1^ to 1040 cm^−1^ are attributed to Si-OH deformation vibration [28,29]. The FTIR spectrum of P(MAA-co-EDGMA) displays the characteristic absorption band of C=O vibrations at 1525 cm^−1^ and 1730 cm^−1^, which appear to be from the carbonyl groups in the MAA component; this spectrum is more intense as compared to that of the P(MAA-co-EGDMA)@SiO_2_. In addition, the absorption peaks at 1152 cm^−1^ and 2900–3000 cm^−1,^ which also appear in both spectra, are attributed to the asymmetrical stretchings of C-O and -CH_2_- in the MAA units, respectively. Moreover, the peak at 750 cm^−1^, which is observed in both spectra, is ascribed to the -CH_2_- deformation vibration. Finally, the peak at 505 cm^−1^ is due to the C-C=O in-plane deformation vibration (found in both spectra) [8] (Table 1).

The existence of the inorganic shell was also confirmed through a thermal behavior evaluation of the core–shell SAPs. Taking into account the TGA curve of the P(MAA-co-EGDMA)@SiO_2_ SAPs (Figure 5b), it may be remarked that the first degradation process (30–150 °C, 18 wt.%) is attributed to the loss of physically absorbed water molecules (through the formation of intra- and inter-molecular anhydride links) and acetonitrile (derived from the synthetic procedure), as well as to the chemisorbed water (the H_2_O monolayer formed onto the inorganic shell). The second weight loss (200–550 °C, 34 wt.%) is ascribed to the polymer decomposition. The residual mass at 550 °C corresponds to the SiO_2_ inorganic shell and is equal to approximately 48 wt.% of the material’s initial mass [9,10]. On the other hand, the TGA curve of the P(MAA-co-EGDMA) SAPs demonstrates that the organic core is completely decomposed by 550 °C (Figure 5a).

Figure 6a illustrates the AR(%) results of SAPs immersed in distilled water with a pH value of 5.5 ± 0.1. Considering the diagrams, it can be observed that the values of AR(%) increase as the immersion time elapses for both SAPs. This outcome indicates that the fabricated SAPs retain their property to absorb water for 360 min (6 h). The P(MAA-co-EGDMA) SAPs demonstrated the highest AR(%) after 360 min. Moreover, it can be remarked that during the initial time of immersion in distilled water at pH 5.5 ± 0.1, both SAPs absorb almost the same amounts of water.

The obtained results of the AR(%) tests of SAPs immersed in distilled water with a pH value of 10.0 ± 0.1 are depicted in Figure 6b. It can be seen that the AR(%) values increase for both SAPs as the immersion time elapses, denoting that the fabricated SAPs continue to absorb water. Furthermore, it is observed that the P(MAA-co-EGDMA)@SiO_2_ SAPs demonstrate the highest AR(%) values after 6 h of immersion.

### 3.2. Evaluation of Coatings

Concentrations of 1 wt.% and 5 wt.% SAPs in the binder Edolan AB were used to make coatings on the ABS and ABS/PC plastic plaque substrates. Scanning electron microscopy was used to evaluate the swelling of SAPs due to the steam as a result of the debonding process.

Moreover, in order to confirm the presence and incorporation of SAPs into the coatings during the application process, the coated samples were fractured after being frozen for 20 min in liquid nitrogen [30]; the morphology of the corresponding cross-section of the coatings was evaluated via SEM (Figure 7, Figure 8, Figure 9 and Figure 10). Considering the aforementioned SEM images, it can be noted that the thicknesses of the produced coatings ranged from 12 μm to 23 μm. The presence of SAPs in the fractured coatings is clearly seen in Figure 9b and Figure 10b. Furthermore, taking into account the corresponding EDS analyses, it can be observed that the coatings consist of carbon, oxygen, and titanium as the main elements due to the nature of the Edolan AB binder (Figure 7b and Figure 8b). Moreover, the presence of SAPs in the coatings results in a slight increase in the wt.% of Si as compared to the coatings without SAPs (Figure 9c and Figure 10c). This outcome is further confirmed through the corresponding EDS elemental Si mappings, where the increase in the concentration of Si in the coatings can be clearly seen (Figure 9d and Figure 10d).

After drying, the coatings were scratched according to ISO 2409:2007 (5 × 5 grid, 2 mm distance between cuts); they were subsequently treated with steam for 5 min (Figure 11a). Figure 11b,c present indicative visual images of the coating with 5 wt.% SAPs prior to and after the steam treatment. The scratched 5 wt.% SAP coating before the steam treatment is illustrated in Figure 11b, where the space cuts can be clearly seen. After the steam treatment, the coating with 5 wt.% SAPs was delaminated in the area where the steam was applied (Figure 11c). Figure 11d illustrates the coating without SAPs after the steam treatment. It can be seen that the coating was not delaminated. Obtained specimens were observed on the SEM in order to determine if the absorbance of humidity would help in the coating’s debonding (Figure 11).

Figure 12 and Figure 13 demonstrate the SEM surface images of the coatings that were applied onto the ABS and ABS/PC substrates, containing either 1 wt.% SAPs or 5 wt.% SAPs, before and after the steam treatment. Considering the observed images, it may be remarked that higher concentrations of SAPs in the coating lead to greater absorption of humidity/steam. This leads to a higher swelling of the SAPs and, consequently, to the facilitation of the debonding of the coating from the substrate. Additionally, the combination 5 wt.% SAPs with ABS substrate poses the best debonding solution, because of both the higher concentration of SAPs and the low compatibility of the ABS substrate and the Edolan AB binder. It must be noted that the coatings of Figure 12 and Figure 13 are not scratched.

The SEM surface images of the scratched coatings applied onto the ABS and ABS/PC substrates without SAPs, or of those with either 1 wt.% SAPs or 5 wt.% SAPs in Edolan AB, are illustrated in Figure 14 and Figure 15. It can be seen that the scratched coatings without SAPs kept their morphology after the steam treatment for both substrates (Figure 14a and Figure 15a). On the other hand, the incorporation of SAPs into the coatings resulted in the scratched coatings’ debonding from both substrates, confirming the following trigger mechanism (Figure 14b,c and Figure 15b,c). The trigger mechanism for the core–shell hydrophilic SAPs relies on the different abilities of the core and the shell materials to be swollen. Therefore, under the influence of steam, the SAPs can enhance their shape due to the water absorption, which leads to the breaking of the inorganic shell. As a consequence, there is a reduction in the attachment of the coating to the substrate. This phenomenon enables the detachment of the top coating from the corresponding substrate. The executed debonding experiment is a proof of concept, providing the evidence that the SAPs can indeed facilitate the coatings’ debonding process after having absorbed a sufficient amount of humidity.

## 4. Conclusions

Spherical SAPs in the submicron scale were synthesized by the copolymerization of MAA with EGDMA via radical polymerization, followed by chemical treatments that allowed the formation of an inorganic SiO_2_ shell around the organic spherical cores. The fabricated SAPs were evaluated with respect to their absorption ratio. The results revealed that both SAPs retain their ability to absorb water because their corresponding AR(%) values increase as the immersion time elapses. This is the case for both the solution with a pH value of 5.5 ± 0.1 and the solution with a pH value of 10.0 ± 0.1. Moreover, in basic pH solutions, the produced SAPs demonstrated enhanced absorption properties compared to acidic pH solutions.

The fabricated SAPs were incorporated into debondable primers. The trigger mechanism for the core–shell hydrophilic SAPs relies on their ability to be swollen. Therefore, under the influence of steam (moisture), the SAPs enhance their shape due to the water absorption resulting in the reduction of the attachment of the coating to the substrate, enabling the detachment of the top coating from the corresponding substrate. Concerning the debonding of SAPs coatings after the application of steam, it should be noted that this method is a proof of concept, providing evidence that the SAPs can indeed facilitate the debonding process of the coatings, after having absorbed a sufficient amount of humidity. Taking into account the outcome of this study, it may be remarked that the concept of the triggerable mechanism of SAPs as well as the application were formulated to provide potential technology for the utilization of SAPs in the plastic industries and the recycling sector. Regarding future research on this subject, exposing the coatings to humidity and immersing them in water may be considered in order to test their debonding behavior under these conditions.

## Figures and Tables

**Figure 1 polymers-13-01432-f001:**
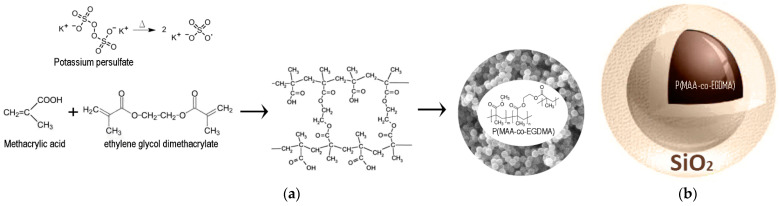
Schematic representation of (**a**) the chemical reactions to produce the organic core of P(MAA-co-EGDMA), and (**b**) the organic core—inorganic shell.

**Figure 2 polymers-13-01432-f002:**
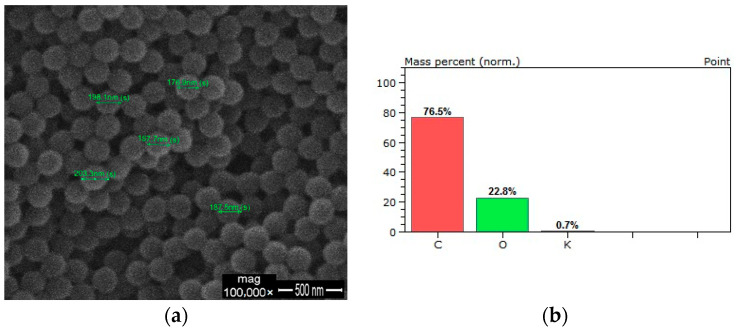
(**a**) SEM image and (**b**) EDS analysis of the produced P(MAA-co-EGDMA) SAPs.

**Figure 3 polymers-13-01432-f003:**
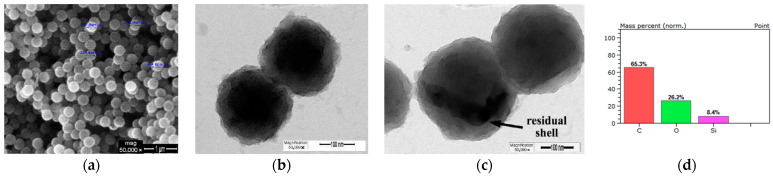
(**a**) SEM image, (**b**,**c**) TEM images, and (**d**) EDS analysis of the produced P(MAA-co-EGDMA)@SiO_2_ core–shell SAPs.

**Figure 4 polymers-13-01432-f004:**
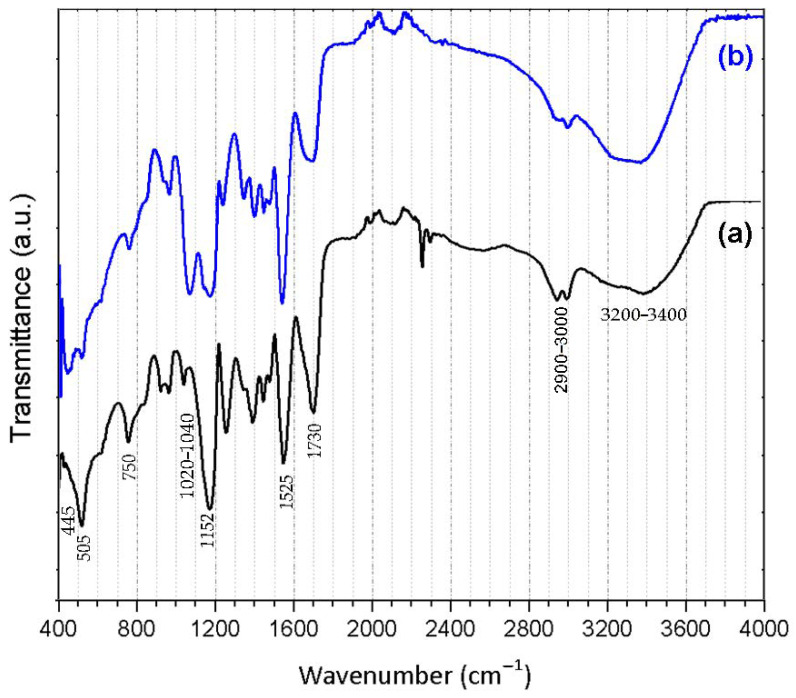
FTIR spectra of the produced (**a**) P(MAA-co-EGDMA) SAPs and (**b**) P(MAA-co-EGDMA)@SiO_2_ SAPs.

**Figure 5 polymers-13-01432-f005:**
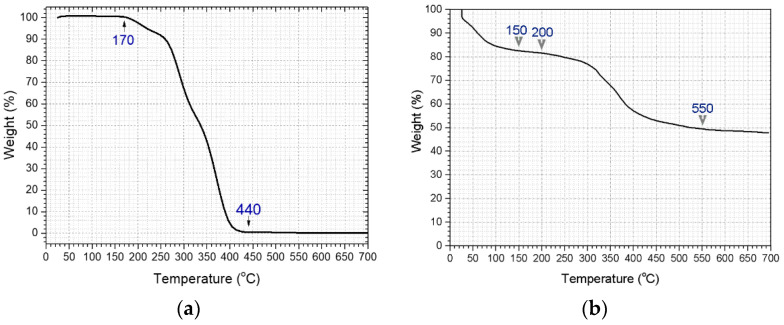
TGA curve of (**a**) P(MAA-co-EGDMA) SAPs and (**b**) P(MAA-co-EGDMA)@SiO_2_ SAPs in nitrogen atmosphere.

**Figure 6 polymers-13-01432-f006:**
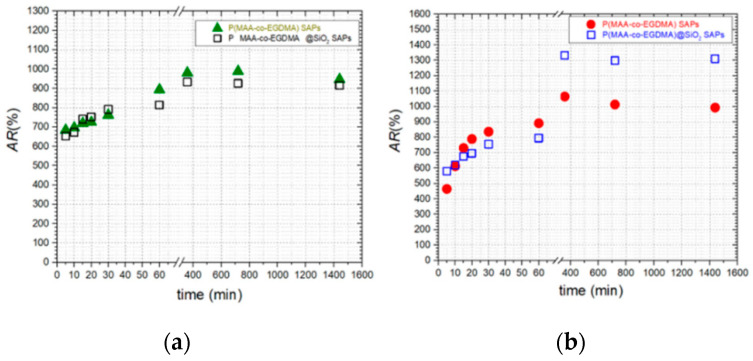
The evolution of AR(%) vs. time of the produced SAPs at (**a**) pH = 5.5 ± 0.1 and (**b**) pH = 10.0 ± 0.1.

**Figure 7 polymers-13-01432-f007:**
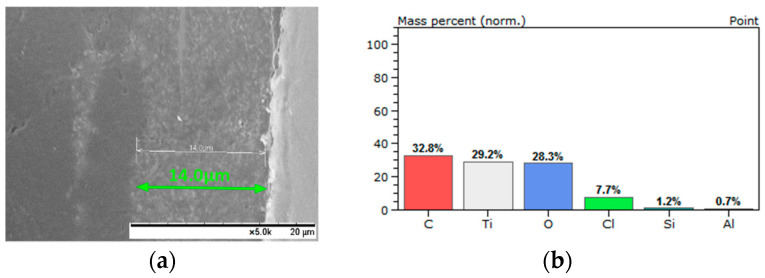
Edolan AB coating on ABS substrate: (**a**) cross-section SEM image and (**b**) EDS analysis.

**Figure 8 polymers-13-01432-f008:**
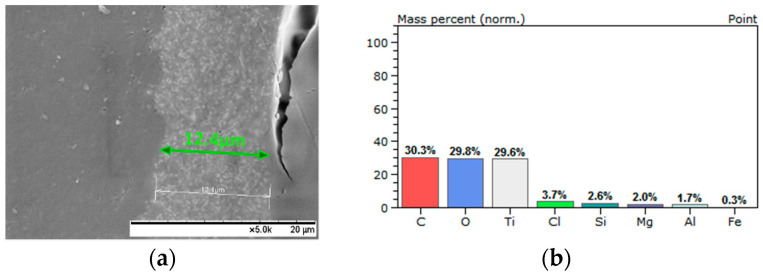
Edolan AB coating on ABS/PC substrate: (**a**) cross-section SEM image and (**b**) EDS analysis.

**Figure 9 polymers-13-01432-f009:**
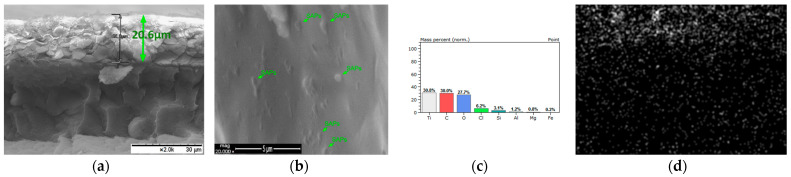
Edolan AB coating with 5 wt.% SAPs on ABS substrate: (**a**) cross-section SEM image magnification ×2000; (**b**) cross-section SEM image magnification ×20,000; (**c**) EDS analysis; (**d**) EDS mapping.

**Figure 10 polymers-13-01432-f010:**
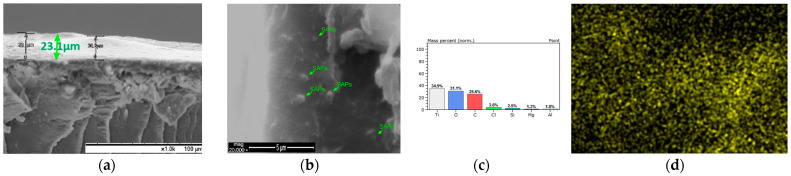
Edolan AB coating with 5 wt.% SAPs on ABS/PC substrate: (**a**) cross-section SEM image magnification ×2000; (**b**) cross-section SEM image magnification ×20,000; (**c**) EDS analysis; (**d**) EDS mapping.

**Figure 11 polymers-13-01432-f011:**
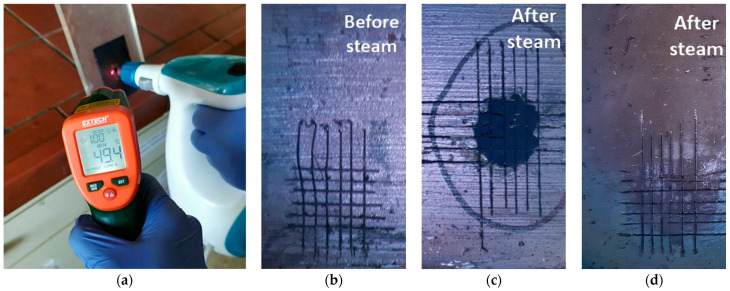
(**a**) Steam treatment process of scratched coating; indicative visual images of scratched coatings on ABS substrates, (**b**) 5 wt.% SAP coating before steam treatment, (**c**) 5 wt.% SAP coating after steam treatment, (**d**) without SAP coating after steam treatment.

**Figure 12 polymers-13-01432-f012:**
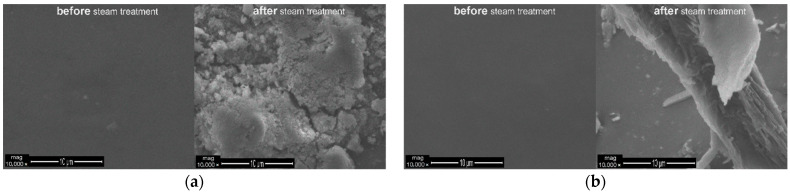
SEM images of the coatings’ surface with: (**a**) 1 wt.% SAPs and (**b**) 5 wt.% SAPs; before and after steam treatment—Edolan AB on ABS substrates.

**Figure 13 polymers-13-01432-f013:**
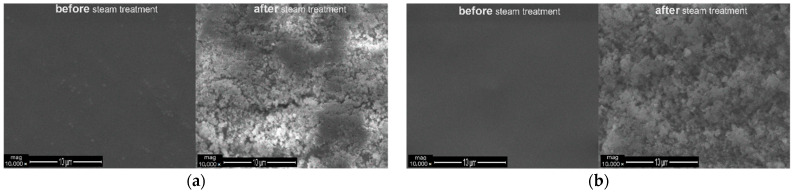
SEM images of the coatings’ surface with: (**a**) 1 wt.% SAPs and (**b**) 5 wt.% SAPs; before and after steam treatment—Edolan AB on ABS/PC substrates.

**Figure 14 polymers-13-01432-f014:**
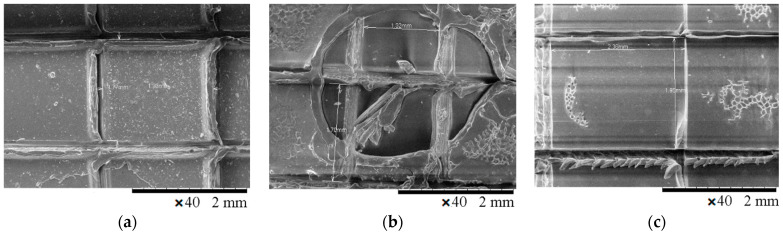
SEM images of the scratched coatings’ surface: (**a**) without SAPs, (**b**) with 1 wt.% SAPs, and (**c**) with 5 wt.% SAPs; after steam treatment—Edolan AB on ABS substrates.

**Figure 15 polymers-13-01432-f015:**
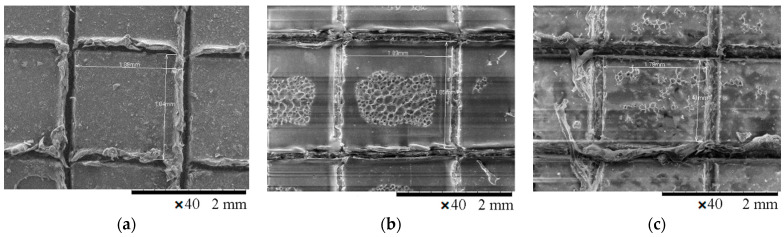
SEM images of the scratched coatings’ surface: (**a**) without SAPs, (**b**) with 1 wt.% SAPs, and (**c**) with 5 wt.% SAPs; after steam treatment—Edolan AB on ABS/PC substrates.

**Table 1 polymers-13-01432-t001:** FT-IR peak analysis.

Wavenumber (cm^−1^)	Corresponding Bond
3200–3400 2900–3000	-OH stretch -CH_2_- asymmetrical stretching
1525, 1730	-C=O vibration
1152	asymmetrical stretching -C-O-
1046, 782, 445	Si-O-Si vibrations
1020–1040	Si-OH
750	-CH_2_- deformation vibration
505	C-C=O in-plane deformation vibration

## Data Availability

Data sharing is not applicable.

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
