# Peer review of "Triggerable Super Absorbent Polymers for Coating Debonding Applications"

_polymers, 2021, doi:10.3390/polym13091432_

Round 1

Reviewer 1 Report

The work describes the synthesis and characterization of core-shell SAPs and their use in a proof-of-concept to promote debonding of coatings applied onto thermoplastic substrates. The topic of reversible adhesion and recyclability is of paramount importance in the context of circular economy, and the use of SAPs in different fields is quite actual, altogether fitting the scope of the journal. Besides, the work is well-structured, and authors used a set of complementary techniques to reach their main conclusions. Therefore, I advise publication of the work.

Please see some minor revisions which may be considered before publication of the manuscript:

1-Chemistry of the Edolan AB primer should be specified in section 2.3.

2-SEM/TEM (Figure 3): Please explain the differences in size for the capsules observed by SEM (around 310 nm) and the capsules observed by TEM (around 150 nm). Furthermore, would it be expected that the SiO2 shell is more or less dense than the SAP core?

3-Figure 4: FTIR of SAPs with and without SiO2 could be overlapped. It would allow an easier comparison of the peaks. Also, include the assignment of bands around 2900-3000 cm-1.

4-TGA curve of SAPs without SiO2 could be added for comparison of the thermal degradation processes, to check if SiO2 promotes further thermal stability of the SAPs.

5-Quantification of Si by EDS (Figures 7-10): There is an increase in Si signal when SAPs are added to coating on ABS substrate, but that increase is not seen for the coating on ABS/PC, which may be due to difficulties in the EDS quantitative analysis for such low concentrations – the SAPs@SiO2 have only 8.4% of Si and only 5wt% of SAPs was used. (increase of 0.4 % in the signal could be expected). Due to really minor variations, EDS mapping could be a better tool as SAPs could be associated with areas of Si signal intensification. In any case, revision of the text should be considered.

6-Figure 11: can picture of the coating without SAPs before and after steam be presented for comparison?

7-What happens if humidity or immersion in water occurs: do the coatings debond or is it occurring only when steam is used? If this has not yet been considered, I advise inclusion of a few words about these points as future work in the conclusions.

Author Response

Dear Editor,

Below follows a detailed list of answers (in red) to the Reviewer’s constructive comments together with the changes that we have made in the manuscript.

The work describes the synthesis and characterization of core-shell SAPs and their use in a proof-of-concept to promote debonding of coatings applied onto thermoplastic substrates. The topic of reversible adhesion and recyclability is of paramount importance in the context of circular economy, and the use of SAPs in different fields is quite actual, altogether fitting the scope of the journal. Besides, the work is well-structured, and authors used a set of complementary techniques to reach their main conclusions. Therefore, I advise publication of the work.

 Please see some minor revisions which may be considered before publication of the manuscript:

The authors would like to thank the Reviewer for his recommendations.

Point 1: Chemistry of the Edolan AB primer should be specified in section 2.3. 

Response 1: Elolan AB composition is not available to the public. Edolan AB primer is a white aqueous dispersion of an anionic acrylic polymer with density 1.0 – 1.1 g/cm³ that is miscible with water. It consists of C, O, Al, Si and Ti elements as evidenced by the EDS analysis.

Point 2: SEM/TEM (Figure 3): Please explain the differences in size for the capsules observed by SEM (around 310 nm) and the capsules observed by TEM (around 150 nm). Furthermore, would it be expected that the SiO2 shell is more or less dense than the SAP core.

Response 2: The SiO2 shell is inorganic, while the core is organic. The polymeric core, due to the bigger molecular weight of the polymeric chains, is denser than the inorganic shell, which can also be witnessed by the thin external layer in the TEM images (Chen et al. Composites Part A: Applied Science and Manufacturing, Volume 133, June 2020, 105832). The differences in size can be assigned to the better distinctive ability of the TEM technique. However, an additional TEM image has been placed in order to further estimate the size of the core-shell SAPs. It is mentioned that in the TEM images the size is ranged from 220-305 nm.

Point 3: Figure 4: FTIR of SAPs with and without SiO2 could be overlapped. It would allow an easier comparison of the peaks. Also, include the assignment of bands around 2900-3000 cm-1.

Response 3: Figure 4 was revised in order the two FT-IR spectra of SAPs with and without SiO2 to be overlapped, so now the comparison of the peaks is easier. The assignment of bands around 2900-3000 cm-1 have been also included; -CH2- asymmetrical stretching.

Point 4: TGA curve of SAPs without SiO2 could be added for comparison of the thermal degradation processes, to check if SiO2 promotes further thermal stability of the SAPs.

Response 4: The TGA curve of SAPs without SiO2 has been added for comparison of the thermal degradation processes.

Point 5: Quantification of Si by EDS (Figures 7-10): There is an increase in Si signal when SAPs are added to coating on ABS substrate, but that increase is not seen for the coating on ABS/PC, which may be due to difficulties in the EDS quantitative analysis for such low concentrations – the SAPs@SiO2 have only 8.4% of Si and only 5wt% of SAPs was used. (increase of 0.4 % in the signal could be expected). Due to really minor variations, EDS mapping could be a better tool as SAPs could be associated with areas of Si signal intensification. In any case, revision of the text should be considered.

Response 5: The text has been revised and the corresponding EDS elemental Si mappings have been added (Figure 9d and 10d).

Point 6: Figure 11: can picture of the coating without SAPs before and after steam be presented for comparison?

Response 6: Figure 11 was revised. Moreover, visual image of coating without SAPs after steam treatment was added (Figure 11d).

Point 7: What happens if humidity or immersion in water occurs: do the coatings debond or is it occurring only when steam is used? If this has not yet been considered, I advise inclusion of a few words about these points as future work in the conclusions.

Response 7: The coatings have not examined with respect to their debonding behaviour after immersion in water. This set of experiments has been included as future works in the conclusions.

Reviewer 2 Report

Spherical super absorbent polymers (SAPs) in the submicron scale were synthesized by the copolymerization of MAA with EGDMA via radical polymerization, followed by chemical treatments that allow the formation of an inorganic SiO2 shell onto the organic spherical cores. Under the influence of steam, the SAPs can enhance their shape due to the water absorption and breaking of the inorganic shell resulting in the reduction of the attachment between the primer layer and both the top coating and substrate, enabling the detachment of the top coating from the corresponding substrate. These contents are interesting and fall well within the scope of polymer engineering. I have no hesitation to recommend its acceptance for publication in POLYMERS. However, minor revision is needed for an improvement. 1)It should be better to give some quantitative results in your ABSTRACT for the readers to cite your job. 2)It should be better to give several sentences about the core-shell structures, which can be fabricated using a wide variety of methods such as electrospinning (Polymers 2020, 12, 2034), electrospraying (Colloids and Surface B - Biointerfaces, 2021, 201, 111629), molecular self-assembly, and so on. 3)Some new methods about creating core-shell polymeric structures (Polymers 2021, 13, 226) can be discussed in the fabrication section. 4)The quality of Figures can be improved in terms of clearness and easy reading. 5)The scale bars in Figure 12 to 15 can be enlarged for the readers.

Author Response

Dear Editor,

Below follows a detailed list of answers (in red) to the Reviewer’s constructive comments together with the changes that we have made in the manuscript.

Comments and Suggestions for Authors

Spherical super absorbent polymers (SAPs) in the submicron scale were synthesized by the copolymerization of MAA with EGDMA via radical polymerization, followed by chemical treatments that allow the formation of an inorganic SiO2 shell onto the organic spherical cores. Under the influence of steam, the SAPs can enhance their shape due to the water absorption and breaking of the inorganic shell resulting in the reduction of the attachment between the primer layer and both the top coating and substrate, enabling the detachment of the top coating from the corresponding substrate. These contents are interesting and fall well within the scope of polymer engineering. I have no hesitation to recommend its acceptance for publication in POLYMERS. However, minor revision is needed for an improvement.

The authors would like to thank the Reviewer for his recommendations.

Point 1: It should be better to give some quantitative results in your ABSTRACT for the readers to cite your job.

Response 1: Quantitative results have been added in the Abstract.

Point 2: It should be better to give several sentences about the core-shell structures, which can be fabricated using a wide variety of methods such as electrospinning (Polymers 2020, 12, 2034), electrospraying (Colloids and Surface B - Biointerfaces, 2021, 201, 111629), molecular self-assembly, and so on.

Response 2: Several methods related to the fabrication of the core-shell structures have been placed in the manuscript (References 20-23).

Point 3: Some new methods about creating core-shell polymeric structures (Polymers 2021, 13, 226) can be discussed in the fabrication section.

Response 3: New methods about creating core-shell polymeric structures (Polymers 2021, 13, 226) have been placed and discussed in the fabrication section.

Point 4: The quality of Figures can be improved in terms of clearness and easy reading.

Response 4: The quality of the figures has been improved.

Point 5: The scale bars in Figure 12 to 15 can be enlarged for the readers.

Response 5: Figures 12-15 have been revised in order the scale bars to be enlarged.